# Geon3D: Benchmarking 3D Shape Bias towards Building Robust Machine Vision

**Yutaro Yamada**[†], **Yuval Kluger**[‡], **Sahand Negahban**[†], **Ilker Yildirim**[†,▷]

Department of [†]Statistics and Data Science, [‡]Applied Mathematics, [▷]Psychology

Yale University

yutaro.yamada@yale.edu

## Abstract

Human vision, unlike existing machine vision systems, is surprisingly robust to environmental variation, including both naturally occuring disturbances (e.g., fog, snow, occlusion) and artificial corruptions (e.g., adversarial examples). Such robustness, at least in part, arises from our ability to infer 3D geometry from 2D retinal projections—the ability to go from images to their underlying causes, including the 3D scene. How can we design machine learning systems with such strong shape bias? In this work, we view 3D reconstruction as a pretraining method for building more robust vision systems. Recent studies explore the role of shape bias in the robustness of vision models. However, most current approaches to increase shape bias based on ImageNet take an indirect approach, attempting to instead reduce texture bias via structured data augmentation. These approaches do not directly nor fully exploit the relationship between 2D features and their underlying 3D shapes. To fill this gap, we introduce a novel dataset called Geon3D, which is derived from objects that emphasize variation across shape features that the human visual system is thought to be particularly sensitive. This dataset enables, for the first time, a controlled setting where we can isolate the effect of "3D shape bias" in robustifying neural networks, and informs more direct approaches to increase shape bias by exploiting 3D vision tasks. Using Geon3D, we find that CNNs pretrained on 3D reconstruction are more resilient to viewpoint change, rotation, and shift than regular CNNs. Further, when combined with adversarial training, 3D reconstruction pretrained models improve adversarial and common corruption robustness over vanilla adversarially-trained models. This suggests that incorporating 3D shape bias is a promising direction for building robust machine vision systems.

## 1 Introduction

The human visual system recovers rich three-dimensional (3D) geometry, including objects, shapes and surfaces, from two-dimensional (2D) retinal inputs. This ability to make inferences about the underlying scene structure from input images—also known as analysis-by-synthesis—is thought to be critical for the robustness of biological vision to occlusions, distortions, and lighting variations [49, 37, 34]. Current machine vision systems, which emphasize image classification over rich 3D scene inferences, are vulnerable to input noise and transformations. Indeed, state-of-the-art vision models for object classification perform poorly when the images are taken from unrepresentative viewpoints [3]. Moreover, we can construct inputs with slight perturbations that are imperceptible to humans but easily fool machine vision, known as adversarial examples [45]. Such instability not only makes machine learning systems unreliable, but also raises serious security concerns [39, 31]. Existing explanations of why adversarial examples exist focus on finite sample overfitting and

Submitted to the 35th Conference on Neural Information Processing Systems (NeurIPS 2021) Track on Datasets and Benchmarks. Do not distribute.

high-dimensional statistical phenomena [18, 16, 19, 7]. More recently, Ilyas et al. [26] propose "non-robust" features that well-generalize to test data as one of the causes behind adversarial examples. To make matters worse, they empirically show that such features are prevalent in real datasets, and machine vision systems naturally make use of them. This observation implies that unless we pressure the system to avoid exploiting "non-robust" features, adversarial examples will continue to exist. Therefore, for reliable machine visions systems, we must build learning algorithms that inherently emphasize variation that is robust across datasets.

A promising set of candidates to target for robustness is the "causal" variables that underlie the pixel distribution in an image—e.g., the 3D scene structure and how it projects to images. Here we focus on learning features to facilitate inferences about one such causal property, the 3D object shape. In fact, a recent line of work has started to explore methods to increase *shape bias* as a way to make neural network models more robust to image perturbations [17, 46, 47]. A notable example is given by Geirhos et al. [17], who proposes to train a model on Stylized-ImageNet (SIN), which are created by imposing various painting styles to images from ImageNet [13]. However, these approaches are indirect: They attempt to reduce the reliability of texture-related cues in terms of how well they can predict object categories, and then make the assumption that under such a data distribution, the model will instead learn to emphasize shape-related cues in the image. Indeed, Mummadi et al. [35] finds that increased robustness to common corruptions using the SIN approach is not due to increased shape bias, but instead, it arises simply from the data augmentation due to style-variation.Moreover, using ImageNet to study shape bias compounds known confounding factors in this dataset, e.g., the 'photographer bias' (i.e., constrained variability across viewpoints) [2, 3], further complicating inferences about shape bias based on the existing work. For example, existing approaches trained on ImageNet might learn to associate class labels with a limited range of non-textural, surface-related cues such as image contours, but they do not fully or explicitly reflect the relationship between 3D objects and how they are projected to images. Here, we advocate that using controlled data distributions, in terms of both the marginal and joint distributions of texture and shape, is needed to isolate and understand the effect of causal scene variables in the context of robustness.

Thus, to our knowledge, none of the existing approaches directly tested the hypothesis that shape bias—learning representations that enable accurate inferences of 3D from 2D, which we refer to as "3D shape bias"—will induce robustness. Inspired by the robustness of the human vision, our desiderata are that such a robust system should not be easily fooled by naturally occurring challenging viewing conditions (e.g., fog, snow, brightness) nor by artificial image corruptions (e.g., due to adversarial attacks).

In this work, we study whether and to what extent 3D shape bias improves robustness of vision models. To answer this question, we introduce *Geon3D*—a novel, controlled dataset comprised of simple yet realistic shape variations, derived from the human object recognition hypothesis called Geon Theory [5]. This dataset enables us to study 3D shape bias of 3D reconstruction models that learn to represent shapes solely from 2D supervision [36]. We find that CNNs trained for 3D reconstruction are more robust to unseen viewpoints, rotation and translation than regular CNNs. Moreover, when combined with adversarial training, 3D reconstruction pretraining improves common corruption and adversarial robustness over CNNs that only use adversarial training. This suggests that not only can Geon3D be used to measure how shape bias improves robustness, it can also guide the introduction of strong shape bias into machine learning models. Biological vision is not only about knowing what is where, but also about making rich inference about the underlying causes of scenes such as 3D shapes and surfaces [37, 49, 4]. We hope our findings and dataset will aid further studies to build more robust vision models with strong shape bias and encourage the community to tackle robustness problems through the lens of 3D inference and perception as analysis-by-synthesis.

## 2 Approach

We first describe the Geon Theory, which our dataset construction relies on. Next, we explain the data generation process used in the creation of Geon3D (§2.1), and how we train a 3D reconstruction model (§2.2).

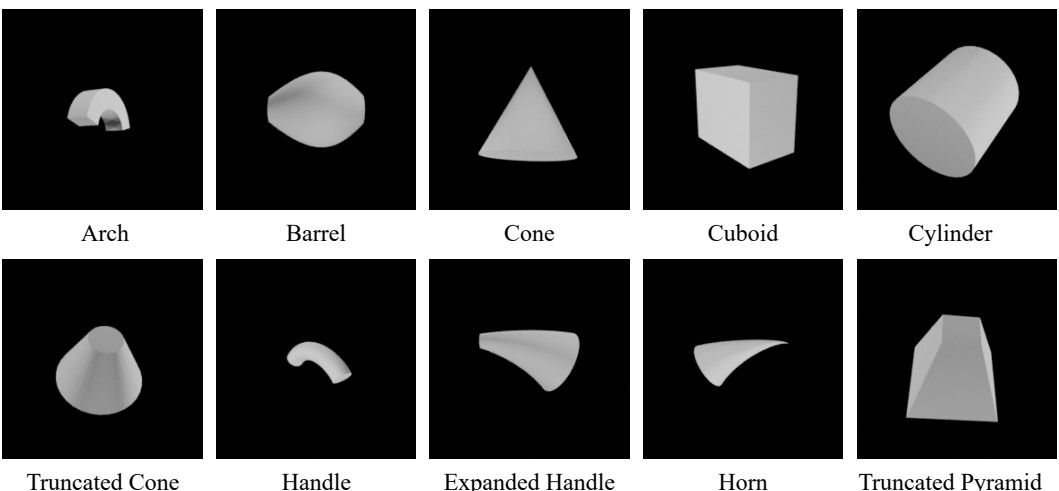

| Arch | Barrel | Cone | Cuboid | Cylinder |
| Truncated Cone | Handle | Expanded Handle | Horn | Truncated Pyramid |

Figure 1: Examples of 10 Geon categories from Geon3D-10. The full list of 40 Geons we construct (Geon3D-40) is provided in the Appendix.

## 2.1 Geon3D Benchmark

The concept of *Geons*—or *Geometric ions*—was originally introduced by Biederman as the building block for his Recognition-by-Components (RBC) Theory [5]. The RBC theory argues that human shape perception segments an object at regions of sharp concavity, modeling an object as a composition of Geons—a subset of generalized cylinders [6]. Similar to generalized cylinders, each Geon is defined by its axis function, cross-section shape, and sweep function. In order to reduce the possible set of generalized cylinders, Biederman considered the properties of the human visual system. He noted that the human visual system is better at distinguishing between straight and curved lines than at estimating curvature; detecting parallelism than estimating the angle between lines; and distinguishing between vertex types such as an arrow, Y, and L-junction [25].

Table 1: Latent features of Geons. S: Straight, C: Curved, Co: Constant, M: Monotonic, EC: Expand and Contract, CE: Contract and Expand, T: Truncated, P: End in a point, CS: End as a curved surface

| Feature | Values |
| --- | --- |
| Axis | S, C |
| Cross-section | S, C |
| Sweep function | Co, M, EC, CE |
| Termination | T, P, CS |

Table 2: Similar Geon categories, where only a single feature differs out of four shape features. "T." stands for "Truncated". "E." stands for "Expanded".

| Geon Category | Difference |
| --- | --- |
| Cone vs. Horn | Axis |
| Handle vs. Arch | Cross-section |
| Cuboid vs. Cyllinder | Cross-section |
| T. Pyramid vs. T. Cone | Cross-section |
| Cuboid vs. Pyramid | Sweep function |
| Barrel vs. T. Cone | Sweep function |
| Horn vs. E. Handle | Termination |

Our focus in this paper is not the RBC theory or whether it is the right way to think about how we see shapes. Instead, we wish to build upon the way Biederman characterized these Geons. Biederman proposed using two to four values to characterize each feature of Geons. Namely, the axis can be straight or curved; the shape of cross section can be straight-edged or curved-edged; the sweep function can be constant, monotonically increasing / decreasing, monotonically increasing and then decreasing (i.e. expand and contract), or monotonically decreasing and then increasing (i.e. contract and expand); the termination can be truncated, end in a point, or end as a curved surface. A summary of these dimensions is given in Table 1.

Representative Geon classes are shown in Figure 1. For example, the "Arch" class is uniquely characterized by its curved axis, straight-edged cross section, constant sweep function, and truncated termination. These values of Geon features are *nonaccidental*—we can determine whether the axis is straight or curved from almost any viewpoint, except for a few *accidental* cases. For instance, an

arch-like curve in the 3D space is perceived as a straight line only when the viewpoint is aligned in a way that the curvature vanishes. These properties make Geons an ideal dataset to analyze 3D shape bias of vision models. For details of data preparation, see Appendix.

## 2.2 3D reconstruction as pretraining

To explore advantages of direct approaches to induce shape bias in vision models, we turn our attention to a class of 3D reconstruction models. The main hypothesis of our study is that the task of 3D reconstruction pressures the model to obtain robust representations.

Recently, there has been significant progress in learning-based approaches to 3D reconstruction, where the data representation can be classified into voxels [11, 41], point clouds [15, 1], mesh [28, 21], and neural implicit representations [33, 10, 40, 44]. We focus on neural implicit representations, where models learn to implicitly represent 3D geometry in neural network parameters after training. We avoid models that require 3D supervision such as ground truth 3D shapes. This is because we are interested in models that only require 2D supervision for training and how inductive bias of 2D-to-3D inference achieves robustness.

Specifically, we use Differentiable Volumetric Rendering (DVR) [36], which consists of a CNN-based image encoder and a differentiable neural rendering module. We train DVR to reconstruct 3D shapes of Geon3D-10. For more details of DVR and 3D reconstruction, we refer the readers to the Appendix.

# 3 Experimental Results

In this section, we demonstrate how 3D shape bias improves model robustness. We evaluate robustness in terms of the Geon3D-10 classification accuracy under various image perturbations. Our 3D-shape-biased classifier is based on the image encoder of the 3D reconstruction model (DVR) that is pretrained to reconstruct Geon3D-10. We add a linear classification layer on top of the image encoder, and then finetune, either just that linear layer (**DVR-Last**) or the entire encoder (**DVR**), for Geon3D-10 classification. Notice that the inputs to all models during classification are only RGB images. (Camera matrices are only used for the rendering module during pretraining for 3D reconstruction.) Our baseline is a vanilla neural network (**Regular**) that is trained normally for Geon3D-10 classification. To see the difference between 3D shape bias and 2D shape bias in the sense of [17], we also evaluate the following models, which are hypothesized to rely their prediction more on shape than texture. **Stylized** is a model trained on Stylized images of Geons. We follow the same protocol as [17] by replacing the texture of each image of Geon3D-10 by a randomly selected texture from paintings through the AdaIn style-transfer algorithm [24]. **Adversarially trained network** (**AT**) is a network that uses adversarial examples during training [32]. Through extensive experiments, Zhang and Zhu [50] demonstrate that AT models develop 2D shape bias, which is considered to explain, in part, the strong adversarial robustness of AT models. In our experiments, we use $L_\infty$ and $L_2$ based adversarial training. **InfoDrop** [43] is a recently proposed model that induces 2D shape bias by decorrelating each layer's output with texture. The method exploits the fact that texture often repeats itself, and hence is highly correlated with and can be predicted by the texture information in the neighboring regions, whereas shape-related features such as edges and contours are less coupled at the locality of neighboring regions. To control for variation in network architectures, we use ImageNet-pretrained ResNet18 for all models we tested. The image encoder of DVR is also initialized using ImageNet-pretrained weights before training for 3D reconstruction of Geons.

**Background variations**   To quantify the effect of textures, we prepare three versions of Geon3D-10: black background, random textured background (Geon3D-10-RandTextured), and correlated background (Geon3D-10-CorrTextured). For Geon3D-10-RandTextured, we replace each black background with a random texture image out of 10 texture categories chosen from the Describable Textures Dataset (DTD) [12].For Geon3D-10-CorrTextured, we choose 10 texture categories from DTD and introduce spurious correlations between Geon category and texture class (i.e., each Geon category is paired with one texture class). Examples of Geon3D with textured background are shown in Figure 3 (Right). These three versions of our dataset allow us to analyze more realistic image conditions as well as to test robustness despite variation and distributional shifts in textures.

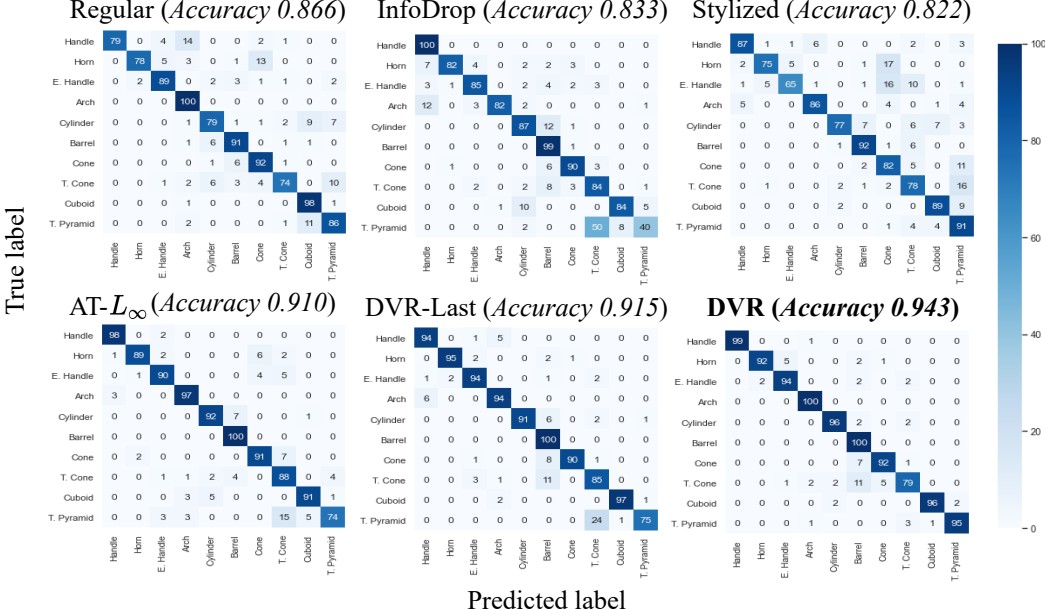

Figure 2: Accuracy per Geon category under unseen viewpoints. Even though all models perform reasonably well, there is still a range of overall accuracy values. In addition, we see that when networks make a mistake, it is often between similar Geon categories (see Table 2 for a list of similar Geon categories). Regular: a baseline model; InfoDrop: a shape-biased model; AT: adversarially trained; Stylized: a network trained on "stylized" version of Geon3D; DVR: We use pretrained weights of the image encoder of Differentiable Volumetric Rendering (3D reconstruction model), a 3D reconstruction model, and finetune all of its layers on the Geon3D-10 classification task. DVR-Last refers to the version where we finetune only the last classification layer.

## 3.1 3D shape bias improves generalization to unseen views and reduces similar category confusion

One of the crucial but often overlooked examples of 3D shape bias that human vision has is "visual completion" [38], which refers to our ability to infer portions of surface that we cannot actually see. For instance, when we look at the top-left image in Figure 3, we automatically recognize it as a whole cube, even though we cannot see its rear side. We view the task of 3D reconstruction as a way to build such an ability into neural networks. In this section, we investigate how such 3D shape bias of DVR improves classification of similar Geon categories under unseen viewpoints, testing both DVR (where we finetune all layers of the image encoder) and DVR-Last (where we finetune only the top classification layer of the image encoder).

The results of per-category classification are shown in Figure 2. We say two Geons are similar when there is only a single shape feature difference, as summarized in Table 2. We see that networks often misclassify similar Geon categories. The vanilla neural network (Regular) often misclassifies "Cone" vs. "Horn", "Handle" vs. "Arch", "Cuboid" vs. "Truncated pyramid", as well as "Truncated cone" vs. "Truncated pyramid".The Geon pairs the InfoDrop model misclassifies include: "Arch" vs. "Handle", "Cyllinder" vs. "Barrel", "Cuboid" vs. "Cyllinder" and "Truncated pyramid" vs. "Truncated cone", which are all pairs with single shape feature difference.

Notably, the Stylized model, which is hypothesized to increase bias towards shape-related features, makes a number of mistakes for similar Geon classes (i.e. "Horn" vs. "Cone", "Cone" vs. "Truncated pyramid", and "Truncated cone" vs. "Truncated pyramid"), similar to the Regular model. This result is consistent with the finding that the Stylized approach [17] does not necessarily induce proper shape bias [35].

AT-$L_\infty$ and DVR-Last perform better than the models listed above, yet still struggle to distinguish "Truncated Pyramid" from "Truncated Cone", where the difference is whether the cross-section is curved or straight (see Table 2). On the other hand, DVR successfully distinguishes these two categories. This shows that 3D pretraining before finetuning for the task of classification facilitates

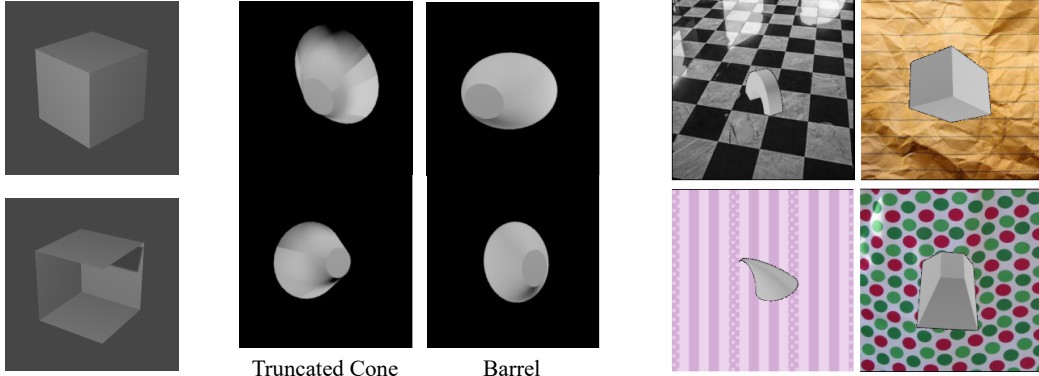

Figure 3: (Left) We humans recognize the top image as a whole cube, automatically filling in the surfaces of its rear, invisible side, although, in principle, there are infinitely many scenes consistent with the sense data , one of which is shown in the bottom image [38]. This illustrates that certain shapes are more readily perceived by the human visual system than others. (Middle) Examples of "Truncated Cone" that are misclassified as "Barrel" by DVR, next to "Barrel" exemplars shown at similar viewpoints.(Right) Example images from Geon3D-10 with textured backgrounds.

recognition of even highly similar shapes. The hardest pair for DVR is "Truncated cone" vs. "Barrel", but the errors the model make appear sensible (Figure 3, middle panel): For example, when the camera points at the smaller side of the "Truncated Cone", then there is uncertainty whether the surface extends beyond self-occlusion by contracting (which would be consistent with the "Barrel" category) or the surface ends at the point of self-occlusion (which would be consistent with the category "Truncated Cone"). Indeed, when we inspected the samples of "Truncated Cone" misclassified as "Barrel" by DVR, we found that for half of those images, the larger side of "Truncated Cone" was self-occluded. Future psychophysical work should quantitatively compare errors made by these models to human behavior.

**Accuracy under rotation and translation (shifting pixels)** CNNs are known to be vulnerable to rotation and shifting of the image pixels [2]. As shown in Table 3, our model (DVR) pretrained with 3D reconstruction performs better than all other models under rotation and shift even though it is not explicitly trained to defend against those attacks. We observe that DVR-Last performs second best, indicating that this "for free" robustness to rotation and shift is largely in place even when finetuning on the classification task is restricted to only linear decoding of the categories.

Table 3: Accuracy of shape-biased classifiers against rotation and shifting of pixels on Geon3D under unseen viewpoints. We randomly add rotations of at most $30°$ and translations of at most 10% of the image size in each $x, y$ direction. We report the mean accuracy and standard deviation over 5 runs of this stochastic procedure over the entire evaluation set.

|  | REGULAR | INFODROP | STYLIZED | AT-$L_2$ | AT-$L_\infty$ | DVR-LAST | DVR |
|---|---|---|---|---|---|---|---|
| ROTATION | $82.18_{(1.06)}$ | $80.76_{(0.69)}$ | $78.47_{(0.57)}$ | $87.00_{(0.57)}$ | $89.58_{(0.48)}$ | $90.44_{(0.30)}$ | $\mathbf{93.46}_{(0.44)}$ |
| SHIFT | $72.28_{(0.43)}$ | $71.86_{(0.63)}$ | $61.44_{(0.29)}$ | $53.84_{(0.71)}$ | $61.50_{(1.11)}$ | $73.24_{(0.73)}$ | $\mathbf{76.52}_{(0.89)}$ |

## 3.2 Robustness against Common Corruptions

In this section, we show that, when combined with adversarial training, 3D pretrained models (denoted as DVR+AT-$L_2$ and DVR+AT-$L_\infty$) improve robustness against common image corruptions, above and beyond what can be accomplished just using adversarial training. For these models, we use adversarial training during the finetuning of the 3D reconstruction model for the Geon3D-10 classification task. Here we evaluate the effect of 3D shape bias not only in the somewhat sterile scenario of the clean, black background images, but also using the background-textured versions of our dataset. To do this, we train all models using Geon3D-10-RandTextured, where we replace the black background with textures randomly sampled from DTD (see Figure 3, right panel, for examples). During evaluation, we use unseen viewpoints.

The results are shown in Table 4. We see that starting adversarial training from DVR-pretrained weights improves robustness across all corruption types, over what can be achieved by only either AT-$L_2$ or AT-$L_\infty$. DVR-AT and AT models fail on "Contrast" and "Fog". This has been a known issue for AT [18], which requires future work to explore. While Stylized performs best under certain corruption types, we can see that DVR-AT-$L_2$ leads to broader robustness across the corruptions we considered.

Table 4: Accuracy of classifiers against common corruptions under unseen viewpoints. All models are trained and evaluated on Geon3D-10 with random textured background. Pretraining on 3D shape reconstruction using DVR leads to broader robustness relative to other models.

|  | REGULAR | INFODROP | STYLIZED | AT-$L_2$ | AT-$L_\infty$ | DVR+AT-$L_2$ | DVR+AT-$L_\infty$ |
|---|---|---|---|---|---|---|---|
| INTACT | 0.741 | 0.596 | 0.701 | 0.691 | 0.464 | **0.758** | 0.513 |
| PIXELATE | 0.608 | 0.458 | 0.653 | 0.623 | 0.415 | **0.719** | 0.470 |
| DEFOCUS BLUR | 0.154 | 0.152 | 0.402 | 0.490 | 0.298 | **0.605** | 0.349 |
| GAUSSIAN NOISE | 0.222 | 0.465 | 0.601 | 0.555 | 0.412 | **0.701** | 0.470 |
| IMPULSE NOISE | 0.187 | 0.270 | 0.497 | 0.322 | 0.136 | **0.594** | 0.148 |
| FROST | 0.144 | 0.269 | **0.638** | 0.142 | 0.209 | 0.148 | 0.240 |
| FOG | 0.338 | 0.281 | **0.659** | 0.187 | 0.120 | 0.264 | 0.130 |
| ELASTIC | 0.427 | 0.314 | 0.428 | 0.416 | 0.266 | **0.499** | 0.307 |
| JPEG | 0.414 | 0.422 | 0.634 | 0.629 | 0.434 | **0.731** | 0.484 |
| CONTRAST | 0.408 | 0.286 | **0.673** | 0.141 | 0.120 | 0.179 | 0.135 |
| BRIGHTNESS | 0.525 | 0.518 | **0.702** | 0.500 | 0.388 | 0.549 | 0.429 |
| ZOOM BLUR | 0.334 | 0.238 | 0.560 | 0.518 | 0.327 | **0.639** | 0.378 |

## 3.3 Robustness to Distributional Shift in Backgrounds

In this section, we evaluate network's robustness to distributional shift in backgrounds. To do this, we train all the models on Geon3D-10-CorrTextured, where we introduce spurious correlation between textured background and Geon category. Therefore, during training, a model can pick up classification signal from both the shape of Geon as well as background texture. To evaluate trained models for background shift, we prepare a test set that breaks the correlation between Geon category and background texture class by cyclically shifting the texture class from $i$ to $i+1$ for $i = 0, ..., 9$, where the class 10 is mapped to the class 0. This is inspired by [17], where they create shape-texture conflicts to measure 2D shape bias in networks trained for ImageNet classification. However, in our case, distributional shift from training to test set is designed to isolate and better measure shape bias by fully disentangling the contributions of texture and shape.

The results are shown in Table 5. We see that 2D shape biased models all perform worse than the 3D shape-biased model (DVR+AT-$L_\infty$). Combining AT with 3D pretraining improves classification accuracy more than 10 % with respect to the best performing variant of AT.

Interestingly, comparing randomized vs. correlated background experiments reveals a stark difference between the two commonly used perturbations in adversarial training ($L_2$ vs. $L_\infty$). Unlike our analysis with uncorrelated, randomized backgrounds, we find that adversarial training using $L_2$ norm completely biases the model towards texture (no apparent shape bias) when such spurious correlation between texture and shape category exists.

Table 5: Accuracy of shape-biased classifiers against distributional shift in backgrounds. Here, all models are trained on Geon3D-10-CorrTextured (with background textures correlated with shape categories) and evaluated on a test set where we break this correlation. See Appendix for results using other common corruptions, where we find DVR+AT-$L_\infty$ provides broadest robustness across the corruptions we tested.

| REGULAR | INFODROP | STYLIZED | AT-$L_2$ | AT-$L_\infty$ | DVR+AT-$L_2$ | DVR+AT-$L_\infty$ |
|---|---|---|---|---|---|---|
| 0.045 | 0.121 | 0.268 | 0.015 | 0.311 | 0.219 | **0.439** |

## 3.4 3D Pretraining Improves Adversarial Robustness

In this section, we show that 3D pretrained AT models improve adversarial robustness over vanilla AT models. We attack our models using $L_\infty$-PGD [32], with 100 iterations and $\epsilon/10$ to be the stepsize,

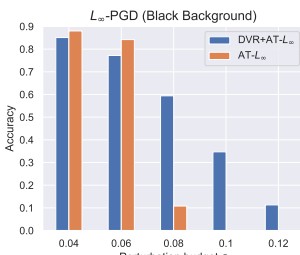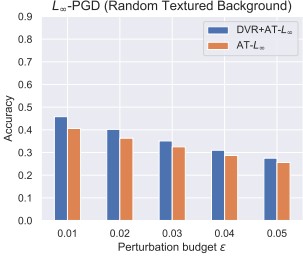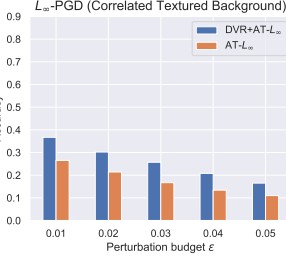

Figure 4: Robustness comparison between AT-$L_\infty$ and DVR+AT-$L_\infty$ with increasing perturbation budget $\epsilon$ on three variations of Geon3D-10. We use $L_\infty$-PGD with 100 iterations and $\epsilon/10$ to be the stepsize. See Appendix for AT-$L_2$ results, where we also find that 3D pretraining improves vanilla AT models.

where $\epsilon$ is the perturbation budget. We compare AT-$L_\infty$ and DVR+AT-$L_\infty$ for black, randomly textured, and correlated textured backgrounds. The results are shown in Figure 4. In the black background set, while 3D pretrained AT slightly performs worse than vanilla AT for smaller epsilon values, it significantly robustifies AT-trained models for large epsilons. A small but appreciable gain in robustness can be seen for the other two backgrounds types. These pattern of results are consistent across attack types, with DVR providing significant robustness over vanilla AT under the $L_2$ regime (see Appendix).

### 3.5 How important is 3D inference?

In this section, we investigate the importance of causal 3D inference to obtain good representations. That is, we explore the impact of having an actual rendering function constrain the representations learned by a model. Our goal in this section is not to further evaluate the robustness of these features, but to measure the efficiency of representations learned under the constraint of a rendering function for the basic task of classification.

To isolate this effect, we compare DVR to Generative Query Networks (GQN) [14]—a scene representation model that can generate scenes from unobserved viewpoints—on novel exemplars from the Geon3D-10 dataset, but using views seen during training. The crucial difference between DVR and GQN is that GQN does not model the geometry of the object explicitly with respect to an actual rendering function. Therefore, the decoder of GQN, which is another neural network based on ConvLSTM, is expected to learn rendering-like operations solely from an objective that aims to maximize the log-likelihood of each observation given other observations of the same scene as context. To control for the difference of network architecture, we train DVR using the same image encoder architecture as GQN, since when we used ResNet18 as an image encoder, GQN did not converge.

Examples of generated images of Geons from GQN are shown in Figure 5 (Left). As we can see, GQN successfully captures the object from novel viewpoints.

To assess the power of representations learned by GQN in the same way as DVR, we take the representation network and add a linear layer on top. We then finetune the linear layer on 10-Geon classification, while freezing the rest of the weights. We compare this model to the architecture-controlled version of the DVR-Last model.

Since GQN can take more than one view of images, we prepare 6 models that are finetuned based on either of $\{1, 2, 4, 8, 16, 32\}$-views. The resulting test accuracy of finetuned GQN encoders against the number of views is shown in Figure 5 (Right). Despite the strong viewpoint generalization of GQN, we see that finetuned GQN requires more than 2 views (i.e., 3 or 4 views) to reach the DVR level accuracy, and only outperforms DVR after we feed more than 8 views. This suggests that the inductive bias from 3D inference is more efficient to obtain good representations.

## 4 Related Work and Discussions

**3D datasets**. Inspired by the success of ImageNet, there have been efforts to create large-scale datasets for 3D vision tasks. ShapeNet [8] provides a large-scale, annotated 3D model dataset. OASIS

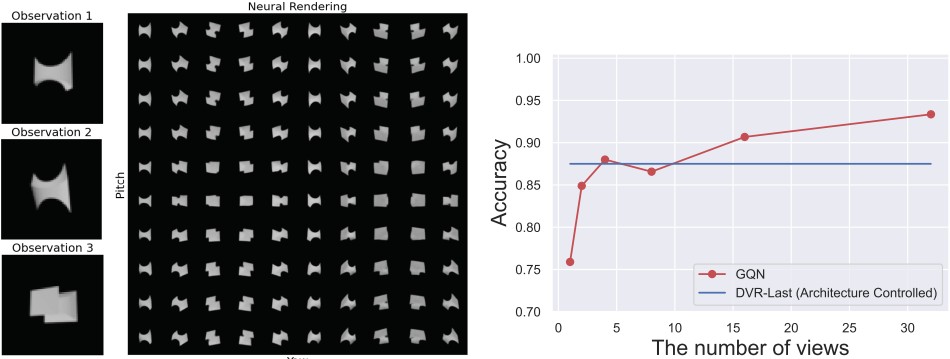

Figure 5: Left: Example Geon images rendered from GQN based on 3 views. Right: GQN Test Accuracy v.s. the number of views. As a reference, we also plot the 1-view DVR accuracy. Here, we used the same architecture for the image encoders of DVR and GQN.

[9] is tailored for tasks of recovering 3D properties from a single-view image, and Rel3D [20] is a benchmark for grounding spatial relations. While these large-scale datasets target 3D vision tasks, Geon3D aims to serve as a diagnostic tool to benchmark how 3D shape bias impacts robustness. Indeed, even though existing learning-based 3D shape reconstruction models can perform well when trained on a single category, these models struggle at multi-category settings (reconstructions become visibly worse when these models are trained on multiple categories of ShapeNet simultaneously). This failure complicates inferences one can make about the role of shape bias in robustness: Is it because the model does not perform well on the reconstruction task to begin with or is it that shape bias has no benefit? As we demonstrate in this work, despite its simplicity relative to these larger datasets, Geon3D reveals that the current vision models struggle with image corruptions and that shape bias induces robustness.

**Part-level robustness vs. Object-level robustness**

To achieve robustness against distributional shifts for complex, real-world objects, we believe it is important to have robust part-whole understanding, which inherently requires understanding of simple geometric objects like Geon3D as a first step. While other 3D datasets such as RotationNet [27] can serve as a testbed for object-level robustness, Geon3D aims to serve as a benchmark for part-level robustness, which is an essential step to achieve object-level robustness. We believe that a simple dataset like Geon3D allows more robustness researchers to explore techniques that are actively being developed in the 3D vision community.

**Analysis-by-synthesis**. Our proposal of using 3D inference to achieve robust vision shares the same goal as analysis-by-synthesis [30, 49, 48]. Given 2D images, these models attempt to find scene parameters such as shape, appearance, and pose, traditionally via top-down stochastic search algorithms like Markov Chain Monte Carlo, and then utilize a graphics engine to reconstruct input. More recently, Efficient Inverse Graphics network (EIG) is proposed [48]. EIG employs a CNN to infer scene parameters of a probabilistic generative model, which is based on a multistage 3D graphics program, and use the aforementioned generative model to synthesize input images. Just like inverse graphics model, such image encoder in 3D reconstruction model has to encode a useful representation for 3D reconstruction. For 3D reconstruction models like DVR, we can consider that scene parameters are implicitly represented in the latent space of the encoder, but importantly, learned with respect to a proper rendering function. Even though previous work considered adversarial robustness of variational autoencoders [42], our study is first to evaluate robustness arising from analysis-by-synthesis type computations under 3D scenes.

**Compositionality and 3D reconstruction**. From the perspective of analysis-by-synthesis approaches, robust recognition of a general complex object should come with the ability to reconstruct it. For such robust recognition, a model needs to learn part-to-whole relationships from images [23, 29] along with each part geometry. We believe that signals from 3D reconstruction can help a recognition model to reliably learn part-to-whole relationships, just like how 3D inference improves robustness. Developing such a 3D inference-based recognition model to compose and analyze complex objects is

an important step towards solving robustness problems of more complex datasets such as ImageNet-C [22] and ObjectNet [3].

# 5 Conclusion

We introduce *Geon3D*—a novel image dataset to facilitate 3D shape bias research in neural network communities. This dataset allows us to study shape bias of a class of 3D reconstruction models that only requires 2D supervision. We demonstrate that CNNs trained for 3D reconstruction improve robustness against viewpoint change and spatial transformation such as rotation and shift. We also study other shape-biased models, and show that not a single model is adequately robust to all corruption types we consider on Geon3D. From a divide-and-conquer perspective, it is desirable to solve robustness problems associated with a simple shape dataset like Geon3D on the way to tackling more complex ones like ImageNet. Finally, we believe that achieving near-perfect robustness on Geon3D is one of the important but simple-to-check conditions that a human-like object recognition system needs to satisfy, as it should operate based on fundamental understanding of the 3D structure of our world.

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
