# OpenReview forum: "Geon3D: Benchmarking 3D Shape Bias towards Building Robust Machine Vision"
_NeurIPS.cc/2021/Track/Datasets_and_Benchmarks/Round1 — Submitted to NeurIPS 2021 Datasets and Benchmarks Track (Round 1)_

### Official Review · Reviewer_EqZ1 · 2021-07-03
**Interesting analysis, limited dataset contribution**

**Rating:** 4
**Confidence:** 3
**Clarity:** I found the paper very clearly written.

**Strengths:**

+ The dataset has a high degree of novelty up to my knowledge.
+ The experiments and analysis are, in my opinion, insightful. They show very clearly the benefits of 3D shape bias in a simple dataset, which is free from distractors appearing in large datasets.

**Weaknesses:**

- The paper is focused on the experiments and the analysis, which are relevant, interesting, while dataset-related aspects are more limited. Although not completely out of scope, I think the paper fits better other calls, to me the dataset is not the central part here, but the experiments: I cannot see a reason why 3D shape bias should be evaluated in the Geon3D dataset and not others, e.g., Kevin Lai's Washington RGB-D objects dataset.

- The simplicity of the generated data, while highlighting some interesting aspects, raises questions on the generalization of the conclusions to more realistic data. In my case, for example, I wonder if the benefits of 3D pre-training still holds for textured objects or complex shapes. The conclusions of the authors are very interesting but they do not explore the limits of their findings, that is, if the conclusions are still valid for more realistic setups.

- The value of the dataset for future uses is unclear. In most of the tasks of the paper, the accuracy of the authors' models is already very high, so its value for developing new approaches might be limited. The authors suggest using it as sanity check, which is quite a limited use, and again, does not ensure that approaches will work in more complicated setups. The authors did not clearly define benchmarks or specific tasks, nor they detail the file structure of the data. Again, in my opinion, the paper is not data-centric and the focus is on the authors' experiments and analysis.

**Additional Feedback:**

My opinion of this work is that the data is not its most relevant part, and the experimental analysis is insightful and promising. I think this track is not the best place for this work. And I also think that the work, although interesting, leaves open questions regarding more realistic images. I would encourage the authors to complete their work by extending the experiments (textured objects, more general backgrounds, real-image datasets) and submit to another venue.

**Correctness:**

The paper is correct up to the extent I checked. The dataset is correctly motivated, and the experiments are clearly explained and correctly analysed.

**Documentation:**

+ The authors provided the URL and I could easily download the dataset.
- There is insufficient detail on the organization of the data (although this is not critical in this case, as the data has a manageable size)
- The authors state their intention to maintain the dataset, but a specific plan is missing.

**Ethics:**

Not applicable for this data.

**Relation To Prior Work:**

The paper references the most related works, up to my knowledge.

**Summary And Contributions:**

The paper introduces the dataset Geon3D, containing images of simple geometric blocks (Geons) generated from a set of high-level shape features. The dataset is motivated by studies on huma perception, and its goal is to study the role of 3D bias in visual recognition. The authors perform extensive experiments that show how networks pre-trained on 3D data improve the performance over those trained directly on 2D images. Specifically, networks with a differentiable neural rendering module show higher accuracy with background variations, better generalization for viewpoints unseen at training time, a higher degree of robustness against standard image corruptions and adversarial attacks.

---

> ### Author Response · Authors · 2021-07-09
> **According to the official website, the scope of the track is not only about novel dataset but also about “systematic analysis of existing systems on novel dataset.” We believe our work satisfies this criteria.**
>
> We thank the reviewer for taking their time and feedback. Below, we respond to the raised points:
>
> > The paper is focused on the experiments and the analysis, which are relevant, interesting, while dataset-related aspects are more limited. Although not completely out of scope, I think the paper fits better other calls, to me the dataset is not the central part here, but the experiments: I cannot see a reason why 3D shape bias should be evaluated in the Geon3D dataset and not others, e.g., Kevin Lai's Washington RGB-D objects dataset.
> > “dataset-related aspects are limited” “this track is not the best place for this work.”
>
> - We appreciate that Reviewer EqZ1 considers our experimental analysis “promising and insightful.” Reviewer EqZ1 notes that it is out of the scope of this track, due to the limited dataset contribution. However, according to the official website, the scope of the track is not only about novel dataset but also about “systematic analysis of existing systems on novel dataset.” We believe our work satisfies this criteria, and thus we don’t think our work is out of the track.
>
> > The simplicity of the generated data, while highlighting some interesting aspects, raises questions on the generalization of the conclusions to more realistic data. In my case, for example, I wonder if the benefits of 3D pre-training still holds for textured objects or complex shapes. The conclusions of the authors are very interesting but they do not explore the limits of their findings, that is, if the conclusions are still valid for more realistic setups.
>
> (Please also see our first comment to Reviewer ZR5f, which has more detailed explanation on this matter.)
> We agree that our findings are focused on the simple setting. However, given the fact that robustness problems are far from being solved, we think a novel simple dataset like Geon3D is a valuable contribution to the robustness community in general, because such a dataset allows more robustness researchers to explore techniques that are actively being developed in the 3D vision community. Regarding how we plan on extending Geon3D to more complex data, please see our next comment.
>
> > The value of the dataset for future uses is unclear. In most of the tasks of the paper, the accuracy of the authors' models is already very high, so its value for developing new approaches might be limited. The authors suggest using it as sanity check, which is quite a limited use, and again, does not ensure that approaches will work in more complicated setups. The authors did not clearly define benchmarks or specific tasks, nor they detail the file structure of the data. Again, in my opinion, the paper is not data-centric and the focus is on the authors' experiments and analysis.
>
> - You note that “most models already perform high accuracy and there are limited future uses expanding on this dataset.” We would like to clarify that none of the models show satisfactory performance under distributional shift, as shown in Table 4, 5 and Figure 4. We think there are a lot of improvements that need to be made from a robustness perspective.
>
> - We want to emphasize that Geon3D is a dataset for the robustness research community (i.e. adversarial robustness, robustness against distributional shift and spatial transformation). We mainly intend to aid incorporating techniques from 3D vision into robustness research.
>
> - Having said that, we are planning to extend Geon3D to study part-whole hierarchy of more complex objects and scenes (i.e. ShapeNet objects can be constructed from Geon3D-40 as parts). [1, 2, 3]. We hope such an extension would be of interest to the 3D vision community. To achieve robustness against distributional shifts for more complex objects, we think it’s critical to have robust part-whole understanding, which inherently requires understanding of simple geometric objects like Geon3D as a first step. We hope that this also answers the question from Reviewer rwKC about the superiority of Geon3D over ShapeNet or ModelNet/RotationNet. That is, we do not claim superiority of Geon3D over ShapeNet or ModelNet, but what we claim is that we should first solve part-level robustness issues before moving onto whole-object robust recognition.
>
> > The authors state their intention to maintain the dataset, but a specific plan is missing.
> - We plan to create a website to have a link to our dataset, which will be hosted by Amazon S3. We also plan to maintain different versions of Geon3D as we extend the dataset to include more complicated objects by combining Geon3D as parts.
>
> [1] Hinton 2021 “How to represent part-whole hierarchies in a neural network”
> [2] Kosiorek et al. 2019 “Stacked Capsule Autoencoders”
> [3] Hinton et al. 2011 “Transforming Auto-encoders”

---

### Official Review · Reviewer_rwKC · 2021-07-03
**Geon3D is a novel dataset but not good enough.**

**Rating:** 4
**Confidence:** 3

**Strengths:**

- This paper build a 2D image dataset with 3D Geons from different viewpoints and background that can make traditional CNN more robust to transformations like rotation and shift.
- This paper exploit Geons to generate the dataset which is simple yet realistic to obtain shape variations.

**Weaknesses:**

- The dataset is built on Geons which is simple to generate a group of shape variations. And there are 10 Geon categories in the dataset. I am concerned whether these Geons can help better understanding categories in ImageNet for 2D and ShapeNet for 3D. Though these simple Geons can easily produce different variations based on latent features of Geons, there is no explicit connection between Geons and objects in real scenes. I'm not sure the dataset could facilitate the understanding of objects in real scenes and contribute to the communities.
- In L151, authors prepare three versions of Geon3D-10: black, random textured and correlated background, to quantify the effect of textures. I can not figure out the meaning of this process since it just changes the background texture as shown in Fig. 3 (Right) while the texture of Geons is not change. In 2D image bias research [1], they change both background and object texture to generate variations. I think that here should consider the effect of textures for both objects and backgrounds can make experiments more convincing.
- I have downloaded they Geon3D dataset, I found that the image of each Geon is a projection from different viewpoints. RotationNet [2] provides a dataset that includes multi-view images of 3D objects on top of  ModelNet. I think this is a similar work and it also includes 3D shape bias. It is better to explain the superiorities of Geons than the whole object for 3D object understanding.
- The dataset lacks of detailed statistics and explicit validation for the correctness of data collection.
- The Geon3D benchmark is vague. There are no explicit metric definitions for evaluating the results on the proposed dataset.

[1] R. Geirhos, P. Rubisch, C. Michaelis, M. Bethge, F. A. Wichmann, and W. Brendel. ImageNet- trained CNNs are biased towards texture; increasing shape bias improves accuracy and robustness. In International Conference on Learning Representations, Sept. 2018.

[2] Asako Kanezaki and Yasuyuki Matsushita and Yoshifumi Nishida. RotationNet: Joint Object Categorization and Pose Estimation Using Multiviews from Unsupervised Viewpoints. Proceedings of IEEE International Conference on Computer Vision and Pattern Recognition (CVPR). 2018.

**Additional Feedback:**

It is interesting that study the Geons to simplify the 3D object understanding. However, the dataset may need to find the relationship between Geons and  complete objects to demonstrate its contributions, which is more meaningful for the community.

The dataset and benchmark should be formed in a more canonical way and include more detailed information.

\* More feedbacks are in comments.

**Clarity:**

This paper is not well written. The proposed dataset and benchmark lack of some key components as mentioned above, which is not up to the demand of this track.

**Correctness:**

The definition of the Geons is explicit, while the dataset lacks of detailed statistics and validation procedure. The benchmark does not have an explicit definition of evaluation metrics.

**Documentation:**

This work provides the dataset link to download the whole dataset. I have downloaded it and the dataset is well-structured. But I do not find the documentation about how to use the dataset.

**Ethics:**

It seems that there are no obvious ethical concerns in Geon3D.

**Relation To Prior Work:**

This paper clearly discusses the differences from 3D datasets, e.g. ShapeNet, OASIS and Rel3D. RotationNet should be compared in the paper as mentioned in weaknesses.


**Summary And Contributions:**

This paper introduces an image dataset for facilitating 3D shape bias research. The dataset is aimed to exploit the relationship between 2D features and 3D shapes with small components of objects. Besides, it emphasizes the variation across shape features. The image generalized from 3D objects via different viewpoints can enlarge the 2D image dataset, which can better generalize to unseen rotation and shift than regular CNN.

---

> ### Author Response · Authors · 2021-07-09
> **Please also consider the value of our experimental analysis from the robustness literature perspective.**
>
> We thank the reviewer for taking their time and feedback on our work. Below we will respond to raised points.
>
> > The dataset is built on Geons which is simple to generate a group of shape variations. And there are 10 Geon categories in the dataset. I am concerned whether these Geons can help better understanding categories in ImageNet for 2D and ShapeNet for 3D. Though these simple Geons can easily produce different variations based on latent features of Geons, there is no explicit connection between Geons and objects in real scenes. I'm not sure the dataset could facilitate the understanding of objects in real scenes and contribute to the communities.
>
> > It is better to explain the superiorities of Geons than the whole object for 3D object understanding.
>
> Please refer to the third comment to Reviewer EqZ1 regarding the connection between Geon3D and more complicated objects, and the usefulness of Geons compared to other whole object datasets. Please also see our first comment to Reviewer ZR5f to see why we think the simple dataset is important for robustness research.
>
> > In L151, authors prepare three versions of Geon3D-10: black, random textured and correlated background, to quantify the effect of textures. I can not figure out the meaning of this process since it just changes the background texture as shown in Fig. 3 (Right) while the texture of Geons is not change. In 2D image bias research [1], they change both background and object texture to generate variations. I think that here should consider the effect of textures for both objects and backgrounds can make experiments more convincing.
>
> - Our aim was to see how much the background has an effect to model robustness. Therefore, if we also add textures to Geons, we cannot isolate the effect of the background shift.
> - In [1], they change both background and object texture because they are interested in how much model prediction is biased towards shape vs. texture, which has a different aim.
>
> > The dataset lacks of detailed statistics and explicit validation for the correctness of data collection.
>
> - If you mean “detailed statistics” by the number of images per category, etc. such information is already in the text and appendix. We will make a table as a summary in the appendix to clarify this point.
> - By “explicit validation for the correctness of data collection” what do you exactly mean?
>
> > The Geon3D benchmark is vague. There are no explicit metric definitions for evaluating the results on the proposed dataset.
>
> - Our metric for benchmarking model robustness is accuracy under different noise types (e.g. Section 3.1, 3.2, 3.3, 3.4). Unless we achieve near-perfect accuracy on each noise type, we don’t think robustness issues are solved on this dataset. We will clarify this point in the revision.

---

> > ### Comment · Reviewer_rwKC · 2021-07-20
> > **The contribution for part-whole understanding should be strengthened.**
> >
> > The part-whole understanding is a novel research are in recent years. And I agree with that it should validate the robustness on a simple part dataset first, then extend to a more complicated scene. Actually, there are part-level 3D datasets in 3D vision, ShapeNet part dataset [1] and PartNet [2]. I believe there exist gaps between Geon3D and these real parts of 3D objects. I suggest that this work is better to validate that shape bias in Geon3D could represent these real parts of 3D objects in [1, 2], like the comments for reviewer EqZ1,
> > > Having said that, we are planning to extend Geon3D to study part-whole hierarchy of more complex objects and scenes (i.e. ShapeNet objects can be constructed from Geon3D-40 as parts).
> >
> > The extension of Geon3D to more complex objects (i.e. ShapeNet, PartNet) could strengthen its contribution for 3D vision community. The authors are recommended to include the extension in the future version of Geon3D and submit it to the 2nd round.
> >
> > [1] Li Yi, Vladimir G. Kim, Duygu Ceylan, I-Chao Shen, Mengyan Yan, Hao Su, Cewu Lu, Qixing Huang, Alla Sheffer and Leonidas Guibas. A scalable active framework for region annotation in 3d shape collections. In SIGGRAPH Asia, 2016.
> >
> > [2] Kaichun Mo, Shilin Zhu, Angel X. Chang, Li Yi, Subarna Tripathi, Leonidas J. Guibas, and Hao Su. PartNet: A large- scale benchmark for fine-grained and hierarchical part-level 3D object understanding. In CVPR, 2019.

---

### Official Review · Reviewer_ZR5f · 2021-07-04
**The dataset promises too much but the effectiveness is uncertain.**

**Rating:** 5
**Confidence:** 2

**Strengths:**

+ The motivation of the paper is to improve the ability to go from images to their underlying 3D shapes, which is a topic worthy of attention. A synthetic dataset is presented to validate this problem.
+ The data generation involves many factors including background texture, viewpoint change, image degradation, distortion, and so on.
+ The design of the experiment is very detailed. Based on the proposed data, the influence of three-dimensional information on the robustness of tasks such as classification is demonstrated.

**Weaknesses:**

- The shapes for building the dataset consist of only dozens of simple basic geometry, which is far from representing the daily structures.  The rendering settings are simple and the images are not photo-real.  These factors determine the huge difference between the generated data and the real data.
- The proposed classification task is too simple to be compared with the classification and recognition task at the level of ImageNet. It is doubtful whether the final test conclusion can be applied to the complex prediction model.
- In terms of a dataset, the scale is small and the composition is simple. I don't think the dataset at this level would be very helpful to the research community.  The article presents meaningful goals to study the role of shape bias in the robustness of vision models, but the scale of the dataset has limited the validation and makes the conclusion less convincing.

**Additional Feedback:**

Considering the limited contribution as the dataset and benchmark, and the fact that the data was built to illustrate the solutions to improve the robustness of vision tasks, I think this paper is not suitable for evaluation on the datasets and benchmarks track. The diversity and rendering photo-reality of the image should be improved to enhance the credibility of the conclusions.

**Clarity:**

The paper is generally well written but there is still room for improvement in language.

**Correctness:**

The correctness of this paper is limited to the experiments of the proposed small data sets and classification tasks, and the correctness of the experimental conclusions is difficult to be confirmed on other complex vision tasks.

**Documentation:**

The documents show sufficient detail about the data generation and the experimental implementation.  The code for rendering using blender is provided.

**Ethics:**

No ethics problems are found in this submission.

**Relation To Prior Work:**

It's a little strange to put the work after the experiment section. In the related works, the introduction of 3D datasets and synthesis based methods is too brief.

**Summary And Contributions:**

The paper presents a novel image dataset to facilitate 3D shape bias research in neural network communities. The dataset is built by rendering designed 3D shapes from different viewpoints, and the data are augmented in various ways. This dataset is used to study the shape bias of a class of 3D reconstruction models that only requires 2D supervision.

---

> ### Author Response · Authors · 2021-07-09
> **Our dataset is mainly for the robustness community, which has a different goal from the general 2D/3D vision community.**
>
> We thank the reviewer for taking their time to give us feedback. We will respond to the raised points below.
>
> > The shapes for building the dataset consist of only dozens of simple basic geometry, which is far from representing the daily structures. The rendering settings are simple and the images are not photo-real. These factors determine the huge difference between the generated data and the real data.
>
> > In terms of a dataset, the scale is small and the composition is simple. I don't think the dataset at this level would be very helpful to the research community. The article presents meaningful goals to study the role of shape bias in the robustness of vision models, but the scale of the dataset has limited the validation and makes the conclusion less convincing.
>
> - We would like to emphasize that our dataset is mainly for the robustness community, which has a different goal from the general 2D/3D vision community.
> - The goal of robustness research is how to ensure robustness when the test distribution is different from the train distribution.
> - As we show in Table 4, 5 and Figure 4, we have not even solved robustness problems in this simple basic geometry setting. If we cannot achieve robustness in this simple rendering setting, how could we expect that current models achieve good robustness on more complex real object data? In the Discussion and Conclusion section, we emphasize that it is necessary to first solve robust object recognition on a simple dataset, which is likely to require radically different architectures and training methods, given how hard it is to ensure robustness against distributional shift, adversarial example, etc with the current set of tools available to us.
> - Please note that it has been 7 years since adversarial examples were first reported, and yet the adversarial robustness problem is currently far from being solved. In terms of robustness, especially adversarial robustness, the current situation has even been described as “pre-Shannon”, referring to how rudimentary security research was before Shannon invented information theory. (Carlini 2019, “On Evaluating Adversarial Robustness” https://www.youtube.com/watch?v=-p2il-V-0fk)
> - To solve these robustness problems, it is likely to require radically different architecture/training methods. We would like to emphasize a simple dataset like MNIST played an important role in the early days of neural network research, and we believe that the simpleness and yet realistic variations of Geon3D are important in this “pre-Shannon” era of robustness research, where robustness researchers can explore techniques from the 3D vision community more easily. We also note that a lot of adversarial robustness papers are still performed on MNIST and CIFAR10, which is far smaller than Geon3D-10. We provide evidence that 3D pretraining can provide further adversarial robustness, which is an novel contribution to adversarial robustness research by itself.
>
> > The proposed classification task is too simple to be compared with the classification and recognition task at the level of ImageNet. It is doubtful whether the final test conclusion can be applied to the complex prediction model.
>
> > The correctness of this paper is limited to the experiments of the proposed small data sets and classification tasks, and the correctness of the experimental conclusions is difficult to be confirmed on other complex vision tasks.
>
> - We agree that our claim is only verified on a small dataset but we note that our 3D shape biased model is based on recent advancement of neural implicit representation. So far, neural implicit representation is only developed for simple texture-less datasets like ShapeNet, and extending neural implicit surface representation to more complex scenes is an active research area. Moreover, neural implicit representation-based 3D reconstruction for multi-category 3D reconstruction is also not achieved with enough accuracy yet. Even though there exist methods to do 3D reconstruction for each object category, this approach generally requires 10 neural network models for 10 object categories. Therefore, we feel that requiring us to verify the correctness of our claim to complex datasets is beyond the scope of a single paper, because it’s still difficult to perform neural implicit representation for complex multi-category datasets using a single network.

---

### Decision · Program_Chairs · 2021-07-26

**Decision:**

Reject

**Comment:**

All reviewers recommended reject. Generally, the reviewers found the dataset to be too simplistic and synthetic and were not convinced about its utility. After reading the authors' responses, the AC agrees with the reviewers.